# Prognosis Prediction Model After Upfront Surgery for Resectable Pancreatic Ductal Adenocarcinoma: A Multicenter Study (OS-HBP-2)

**DOI:** 10.3390/cancers17223694

**Published:** 2025-11-18

**Authors:** Kosei Takagi, Ryuichi Yoshida, Kazuya Yasui, Masayoshi Hioki, Takehiro Okabayashi, Toru Kojima, Yoshikatsu Endo, Daisuke Nobuoka, Kenta Sui, Masaru Inagaki, Susumu Shinoura, Masashi Kimura, Tatsuo Matsuda, Hideki Aoki, Toshiyoshi Fujiwara

**Affiliations:** 1Department of Gastroenterological Surgery, Graduate School of Medicine, Dentistry, and Pharmaceutical Sciences, Okayama University, Okayama 700-8558, Japan; ryu923ichi@gmail.com (R.Y.); pjyv6nvp@s.okayama-u.ac.jp (K.Y.); toshi_f@md.okayama-u.ac.jp (T.F.); 2Department of Surgery, Fukuyama City Hospital, Hiroshima 721-0971, Japan; s2000hioki@yahoo.co.jp (M.H.); nobuoka11074@fchp.jp (D.N.); 3Department of Surgery, Kochi Health Sciences Center, Kochi 781-0111, Japan; takehiro_okabayashi@khsc.or.jp; 4Department of Surgery, Okayama Saiseikai General Hospital, Okayama 700-8511, Japan; trkojima0507@yahoo.co.jp; 5Department of Surgery, Himeji Red Cross Hospital, Hyogo 670-8540, Japan; y-endou@himeji.jrc.or.jp; 6Department of Surgery, Kagawa Prefectural Hospital, Kagawa 760-8557, Japan; suikenta25@ybb.ne.jp; 7Department of Surgery, National Hospital Organization Fukuyama Medical Center, Hiroshima 720-0825, Japan; inagaki.masaru.dp@mail.hosp.go.jp; 8Department of Surgery, Tsuyama Chuo Hospital, Okayama 708-0841, Japan; s.shino@tch.or.jp; 9Department of Surgery, Matsuyama Shimin Hospital, Ehime 790-0067, Japan; kishimrms6717@matsuyama-shimin-hsp.or.jp; 10Department of Surgery, Tenwakai Matsuda Hospital, Okayama 710-0056, Japan; tenwasur@mx5.kct.ne.jp; 11Department of Surgery, National Hospital Organization Iwakuni Clinical Center, Yamaguchi 740-8510, Japan; aoki.hideki.hy@mail.hosp.go.jp

**Keywords:** pancreatic cancer, resectable, upfront Surgery, outcomes

## Abstract

Although several prognostic nomograms have been developed to estimate the prognosis of patients with pancreatic ductal adenocarcinoma (PDAC), further investigation is re-quired to optimize the prognostic assessment tools for patients with resectable PDAC. This multicenter study (*n* = 603) aimed to investigate the prognostic factors for survival and develop a prognosis prediction model after upfront surgery in patients with resectable PDAC. Using the results of the multivariate analyses, a prognosis prediction model for overall survival was constructed. This study suggests that our model may be useful and can be internally validated for predicting overall survival following upfront surgery in patients with resectable PDAC.

## 1. Introduction

Pancreatic ductal adenocarcinoma (PDAC) remains one of the most lethal malignancies with a poor survival rate [1,2]. Currently, the importance of resectability is emphasized when determining the treatment strategies for PDAC [3]. The National Comprehensive Cancer Network guidelines categorize resectability status as resectable, borderline resectable, or unresectable, and propose appropriate workups and treatment options based on patient presentation, cancer type, and disease stage [4]. Upfront surgery (UFS) is recommended for patients with resectable PDAC without high-risk features such as large primary tumors, enlarged regional lymph nodes, and markedly elevated carbohydrate antigen 19-9 (CA 19-9) levels [4]. However, despite recent developments in multidisciplinary treatments for PDAC, prognosis after resection remains poor [5].

Numerous efforts have been made to identify prognostic factors and estimate the prognosis of patients with PDAC [6]. Several prognostic nomograms have been developed and incorporate patient characteristics, tumor factors, and tumor markers [7,8,9]. Moreover, the impact of various nutritional and inflammatory assessment tools, including the modified Glasgow Prognostic Score (mGPS) and prognostic nutritional index, on PDAC has been investigated [10]. However, because the resectability status is evaluated based on radiologic anatomical findings [4], varying degrees of malignancy can be categorized within the same resectability group. Therefore, resectability-specific risk factors should be investigated to optimize prognostic assessment tools for PDAC. Models for predicting early recurrence and preoperative lymph node metastasis have been developed for resectable PDAC [11,12,13]. However, few studies have developed prognostic prediction models for the survival of patients with resectable PDAC.

This multicenter study investigated long-term outcomes and prognostic factors for survival after UFS in patients with resectable PDAC. In addition, a prognostic prediction model was developed. This study was conducted as part of the Okayama Study Group for Hepato-Biliary-Pancreatic Surgery (OS-HBP-2).

## 2. Materials and Methods

### 2.1. Study Cohort

This was a multicenter retrospective study, including 647 patients who underwent UFS for resectable PDAC between January 2013 and December 2017. Clinical data were collected from the OS-HBP [14]. The study was approved by the Ethics Committee of our institution (approval no. 2211-039) and conducted in accordance with the principles outlined in the Declaration of Helsinki. Due to the retrospective nature of this study, the requirement for informed consent was waived.

### 2.2. Clinical Data

Using the anonymous database, we extracted the following data: age, sex, body mass index, biliary drainage, laboratory values including tumor markers, mGPS [15], preoperative radiological findings (tumor location, tumor size, venous involvement, and lymph node metastasis), preoperative clinical findings were classified according to the Unio Internationalis Contra Cancrum (UICC)/American Joint Committee on Cancer (AJCC) seventh edition [16,17], type of procedure (pancreaticoduodenectomy, distal pancreatectomy, or total pancreatectomy), operative factors, pathological factors, including R0 and R1 classification [18]) evaluated by the UICC/AJCC seventh edition [16,17], major complications (Clavien–Dindo classification; ≥grade 3 [19]), postoperative pancreatic fistula (≥grade B [20]), adjuvant chemotherapy, and long-term outcomes (recurrence and status at last follow-up [survival or death]).

Preoperative radiological findings and the initial resectability status were evaluated using computed tomography scans at a multidisciplinary conference. The following criteria for resectable PDAC, based on the National Comprehensive Cancer Network criteria, were used: no tumor contact with major arterial structures (celiac artery, superior mesenteric artery, and/or common hepatic artery) and no tumor contact with the portal vein (PV) or superior mesenteric vein (SMV), with ≤180° contact without vein contour irregularity [4]. Venous involvement was categorized as no tumor contact with the PV or SMV or tumor contact with the PV or SMV of ≤180° in this study [21]. Using C-reactive protein and serum albumin levels, patients were classified as having an mGPS of zero, one, or two [15]. The following cutoff values of CA 19-9 were used: 40 U/mL, normal range; and 500 U/mL, biologically borderline resectable [22]. As CA19-9 levels can be significantly influenced by biliary obstruction, we selected preoperative CA19-9 levels recorded after biliary drainage for this study. Pathological stages were defined using the criteria of the AJCC seventh edition criteria [17]. The definition of R0 resection in this study included the margin edge being free based on the closest margin among all resection margins. Adjuvant chemotherapy included either S-1 or gemcitabine administered 6 months after surgery [23].

### 2.3. Statistical Analysis

All statistical analyses were performed using EZR software (version 1.65; Saitama Medical Center, Jichi Medical University, Saitama, Japan) and the JMP (version 11; SAS Institute, Cary, NC, USA). Categorical variables were presented as proportions, and continuous variables were presented as medians with interquartile ranges (IQRs). Differences between groups were assessed using either Fisher’s exact test or the chi-square test for categorical variables and the Mann–Whitney U test for continuous variables. All reported *p*-values were two-tailed, with an alpha level of 0.05 considered statistically significant.

The overall survival (OS) and recurrence-free survival (RFS) were calculated using the Kaplan–Meier method. Overall survival (OS) was defined as the time interval between surgery and death from any cause, whereas RFS was defined as the time from surgery to recurrence or death from any cause. Univariate and multivariate analyses were performed using the Cox proportional hazards model to identify the prognostic factors for OS. Hazard ratios (HRs) and 95% confidence intervals (CIs) were calculated.

A simple scoring model was developed based on multivariate analyses, with one point assigned to each prognostic factor, demonstrating a similar HR. Thereafter, patients were divided into several groups according to the identified prognostic factors. Another prognostic prediction model was developed to predict the OS using multivariate analysis. Using the area under the receiver operating characteristic curve (AUC), the performance of the model was evaluated. Subsequently, the bootstrap method was used for internal validation to assess the discriminative performance of the model [24]. The predictive validity of the model was evaluated by analyzing the calibration curve and concordance index (C-index).

## 3. Results

### 3.1. Cohort

Among the 647 patients, 603 were included after excluding 44 patients who were Lewis-negative (*n* = 41) and those with postoperative in-hospital mortality (*n* = 3). The characteristics of 603 patients are shown in Table 1. Of 603 patients, 210 (34.8%) underwent biliary drainage. Median CA19-9 level was 111 U/mL (IQR 30–392 U/mL), and preoperative total bilirubin was within the normal range of 0.8 mg/dL (IQR 0.6–1.3 mg/dL).

Tumor locations were the pancreatic head (*n* = 393, 65.2%) and pancreatic body and tail (*n* = 210, 34.8%), with a median tumor size of 2.3 cm (IQR 1.8–3.0 cm). Tumor contact with a PV or SMV of ≤180° was confirmed in 174 (28.9%) patients. The procedures included pancreaticoduodenectomy (*n* = 387, 64.2%), distal pancreatectomy (*n* = 198, 32.8%), and total pancreatectomy (*n* = 18, 3.0%). Regarding pathological findings, 331 (54.9%) patients had lymph node metastases. R0 resection was achieved in 532 (88.2%) patients.

During the follow-up period of 25 months (IQR, 15–38 months), 381 (63.2%) patients experienced recurrence. The median OS and RFS after UFS were 34 months and 15 months, respectively. The 1-, 3-, and 5-year OS rates were 83.7%, 48.2%, and 37.5%, respectively. The 1-, 3-, and 5-year RFS rates were 55.1%, 30.7%, and 27.1%, respectively. The Kaplan–Meier curves for OS and RFS stratified by pathological stage are shown in Figure 1. The 5-year OS rates were 69.7%, 47.1%, and 25.4% for pStages IA and IB and pStage IIA and pStage IIB, respectively (Figure 1a). The 5-year RFS rates were 64.9%, 34.5%, and 15.1% for pStages IA and IB and pStage IIA and pStage IIB, respectively (Figure 1b).

### 3.2. Prognostic Factors Associated with OS

The results of the univariate and multivariate analyses are presented in Table 2. Multivariate analyses identified four independent predictors associated with OS: tumor size > 2 cm (hazard ratio [HR] 1.50, *p* = 0.001), tumor contact with the PV and SMV of ≤180° (HR 1.47, *p* = 0.003), CA 19-9 40–500 U/mL (HR 1.59, *p* = 0.002) and ≥500 U/mL (HR 2.16, *p* < 0.001), and mGPS of two (HR 1.56, *p* = 0.038).

### 3.3. Simple Scoring Model for OS Following UFS

A simple scoring model was developed for all patients, with one or two points assigned to each independent predictor: tumor size > 2 cm (one point), tumor contact with the PV and SMV (one point), CA 19-9 40–500 U/mL (one point) and >500 U/mL (two points), and mGPS of two (one point). The patients were categorized into three groups based on these scores: low risk (0–1 point; *n* = 240), moderate risk (2–3 points; *n* = 313), and high risk (4–5 points; *n* = 50). The clinicopathological characteristics stratified by risk group are shown in Table 3. The high-risk group exhibited the most advanced tumor characteristics. However, no significant differences were observed between the groups in terms of R status, incidence of major complications, or use of adjuvant chemotherapy. The incidence of postoperative recurrence in the low-, moderate-, and high-risk groups was 47.9%, 71.6%, and 84.0%, respectively (*p* < 0.001).

The OS curves for these groups showed overall 3- and 5-year survival rates of 63.8% and 52.4% in the low-risk group, 41.4% and 30.1% in the moderate-risk group, and 16.1% and 12.1% in the high-risk group (log-rank *p* < 0.001) (Figure 2). The OS curves for these groups showed overall 3- and 5-year survival rates of 63.8% and 52.4% in the low-risk group, 41.4% and 30.1% in the moderate-risk group, and 16.1% and 12.1% in the high-risk group, respectively (log-rank *p* < 0.001) (Figure 2).

### 3.4. Prognosis Prediction Model for OS Following UFS

Using the results of multivariate analyses, the predictive probability of the model for OS was constructed (Equation (1); available in Appendix A):S(t∣x) = [S0(t)]^exp(linear predictor)= [exp(−0.008296 * t)]^exp(0.40314 * [tumor size > 2 cm] + 0.39622 * [tumor contact with PV&SMV]  + 0.3962 * [CA 19-9 40–500 or ≥500] + 0.49915 * [mGPS 2])=exp(−0.008296 * t * exp{0.40314 * [tumor size > 2 cm] + 0.39622 * [tumor contact with PV&SMV]  + 0.3962 * [CA 19-9 40–500 or ≥500] + 0.49915 * [mGPS 2])p (predictive probability; %) = S(t∣x) * 100S(t∣x) survival function at time t for an individual with covariates xS0(t) baseline survival function(1)

### 3.5. Model Performance and Calibration

Calibration plots of the predictive model for 1-, 3-, and 5-year OS are shown in Figure 3. The AUCs for 1-, 3-, and 5-year OS were 0.67 (95% CI, 0.61–0.74), 0.67 (95% CI, 0.62–0.72), and 0.68 (95% CI, 0.59–0.76), respectively.

The predicted probability of OS was moderately correlated with the actual likelihood, with a C-index of 0.63 (Figure 4).

### 3.6. A Literature Review

A PubMed literature search was conducted to identify articles that developed nomograms to predict postoperative survival after surgery in patients with PDAC. Table 4 summarizes the 10 relevant articles, including our study [7,8,9,25,26,27,28,29,30]. The study designs included single-center (*n* = 5), multicenter (*n* = 2), and database (*n* = 3) studies. Previously published nomograms targeted PDAC resection. The C-index ranged from 0.63 0.79.

## 4. Discussion

Although UFS has been recognized as the standard of care for patients with resectable PDAC [4], the prognosis remains poor, even in resectable PDAC, probably because of incomplete resection and early postoperative recurrence [31]. In this multicenter retrospective study, we investigated long-term outcomes and preoperative prognostic factors associated with OS after UFS in 603 patients with resectable PDAC. Multivariate analyses identified tumor size, tumor contact with the PV and SMV of ≤180°, CA 19-9 level, and mGPS as independent predictors of OS. Moreover, a prognostic prediction model was developed to estimate OS following UFS for patients with resectable PDAC, and its performance was evaluated using internal validation. To the best of our knowledge, this is the first multicenter study to demonstrate a prognostic prediction model for OS after UFS in patients with resectable PDAC.

Long-term outcomes after UFS in this study, including median OS and RFS of 34 months and 15 months, were relatively better compared with those of previous reports, which reported median OS and RFS of 18.8 (95% CI 12.4–20.6) months and 9 (95% CI 1.6–12.5) months [32]. This improvement might be due to the higher R0 resection and induction rates of adjuvant chemotherapy in this study [32]. However, approximately 60% of patients experienced recurrences. Therefore, further efforts are required to improve the long-term outcomes based on potential risk factors.

Several preoperative factors, including tumor size, tumor contact with the PV and SMV of ≤180°, CA 19-9 level, and mGPS, were identified as prognostic factors associated with OS in this study. Previous studies reporting prognostic nomograms found that tumor size is a predictive parameter for prognosis [7,33]. Our novel finding is that preoperative radiological venous involvement, tumor contact with the PV&SMV of ≤180°, is a prognostic factor for OS. Since venous involvement can be divided into no tumor contact and tumor contact with the PV or SMV [21], we found that no tumor contact and tumor contact with the PV or SMV represented distinctly different risk factors for survival in patients with resectable PDAC. Portomesenteric venous tumor contact is reportedly associated with poor OS, high rates of R1 resection, and lymph node metastasis in patients with resectable PDAC following UFS [34]. CA 19-9 is regarded as an effective predictor of OS, nodal involvement, and margin status after surgery in patients with resectable PDAC [35,36]. A meta-analysis demonstrated that mGPS, a promising inflammatory prognostic indicator, is closely associated with PDAC [37].

Prognostic prediction models for OS were constructed using the results of multivariate analyses. In the simple scoring model, survival curves were stratified according to the risk scores (Figure 2). The postoperative survival probability (%) was calculated using a prognosis prediction model (Equation (1)). Regarding model performance and internal validity, the model’s performance, with an AUC of 0.67 to 0.68, was satisfactory. The calibration plots demonstrated relatively good agreement, with a C-index of 0.63. Because preoperative factors were selected to establish these prognostic prediction systems, we could estimate patient survival preoperatively. Therefore, the potential clinical implications of this model include the survival risk stratification of patients with resectable PDAC.

As shown in Table 3 and Figure 2, the OS and recurrence rates in the moderate- and high-risk groups were significantly lower than those in the low-risk group. Our results suggest that resectable PDAC should not be categorized solely by resectability status but rather subcategorized using various clinical factors, and appropriate multidisciplinary treatments, including neoadjuvant and adjuvant chemotherapy, should be provided. Although the role of neoadjuvant chemotherapy in resectable PDAC remains controversial, our prognostic prediction models may be useful for identifying high-risk groups that may benefit from intensified multidisciplinary care. Considering the growing evidence for neoadjuvant treatments for resectable PDAC, neoadjuvant chemotherapy may improve oncological outcomes and survival, especially in patients with high-risk features of resectable PDAC [38,39,40]. Moreover, evidence for adjuvant therapy for PDAC is well established [23,41,42,43]. The primary rationale behind the use of neoadjuvant and adjuvant therapies in pancreatic cancer is to target and destroy occult micrometastatic disease, which is believed to be the leading cause of early and frequent disease recurrence after curative surgery [44]. Accordingly, a perfect clinical course, including neoadjuvant chemotherapy, surgery, and adjuvant therapy, may be the ultimate protocol for improving outcomes, especially in moderate- and high-risk groups [40].

Several novel findings of our model compared to previous nomograms are summarized in Table 4. As previous models targeted resected PDAC, our model focused only on patients with resectable PDAC. The prognostic factors for survival differ depending on the resectability status. Therefore, the resectability-specific prediction model was reasonable. Moreover, existing nomograms include postoperative parameters such as pathological findings that cannot be assessed preoperatively. In contrast, our model included only preoperative factors. Therefore, we could estimate patient survival preoperatively and construct treatment strategies depending on the risk categories.

The present study had several limitations. Although this was a multicenter study with a relatively large sample size, its retrospective nature may have introduced a potential selection bias in surgical decision-making across centers and limited the causality of the findings. Using multivariate analysis, we identified independent preoperative predictors associated with OS. However, other confounding or residual confounding factors may exist. The discriminative performance of the prognostic prediction model was investigated using internal validation. However, the moderate performance of the prediction model may be a limitation. Moreover, an external validation was not performed to verify the reliability and generalizability of the model. Because external validation is crucial to ensure the model’s applicability in different populations, the lack of external validation of the prediction model is a significant limitation. Future studies should conduct external validations to confirm the applicability of this model.

## 5. Conclusions

The present study demonstrated long-term outcomes and identified preoperative prognostic factors of survival, including tumor size, tumor contact with the PV and SMV of ≤180°, CA 19-9 level, and mGPS, in patients who underwent UFS for resectable PDAC. Moreover, we developed a prognostic prediction model to estimate the OS after UFS for resectable PDAC. Our model may be useful and internally validated for predicting OS after UFS. Intensive multidisciplinary treatment may be required in high-risk patients.

## Figures and Tables

**Figure 1 cancers-17-03694-f001:**
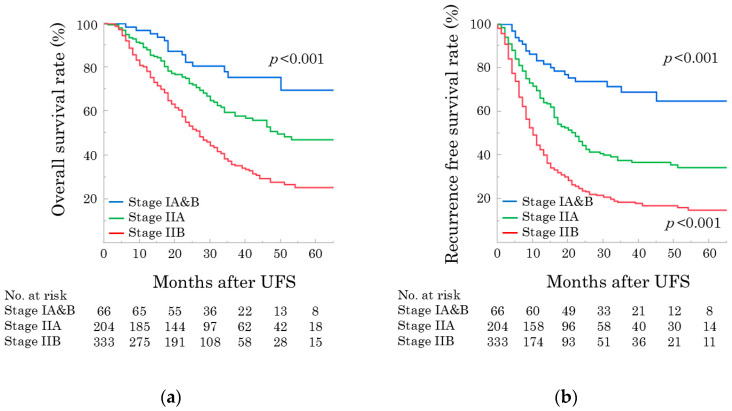
(**a**) Overall survival and (**b**) recurrence-free survival curves after pancreatectomy, stratified by pathological stage. UFS, upfront surgery.

**Figure 2 cancers-17-03694-f002:**
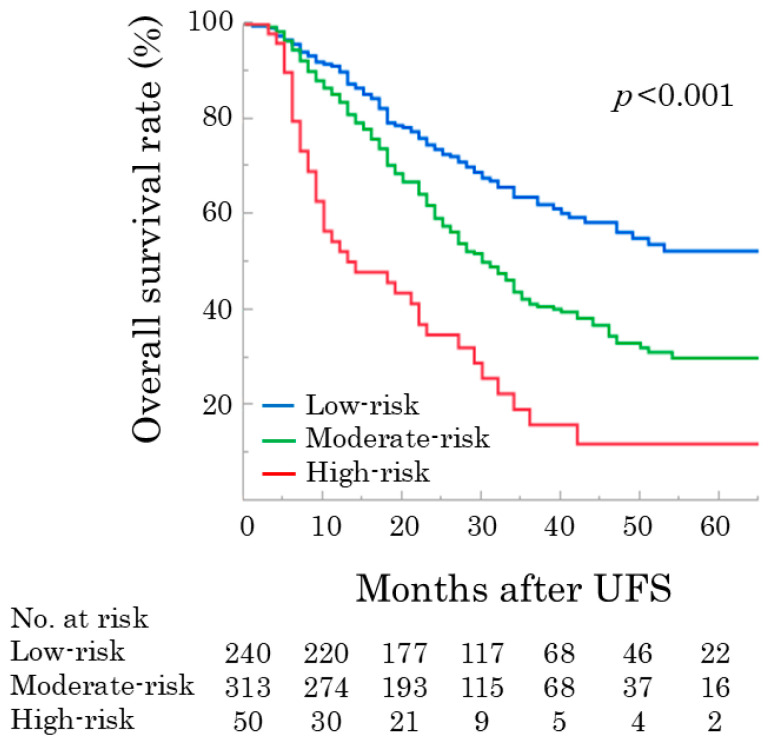
Overall survival curves after upfront surgery for patients in the low-risk, moderate-risk, and high-risk groups (log-rank *p* < 0.001). UFS, upfront surgery.

**Figure 3 cancers-17-03694-f003:**
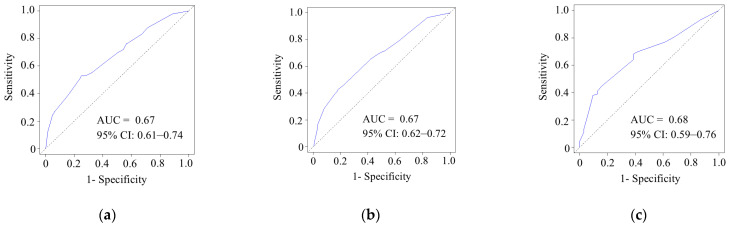
Receiver operating characteristic curves and area under the curve for the model predicting (**a**) 1-year, (**b**) 3-year, and (**c**) 5-year overall survival. AUC, area under the curve; CI, confidence interval. Blue line, observed receiver operating characteristic curve; black line, diagonal.

**Figure 4 cancers-17-03694-f004:**
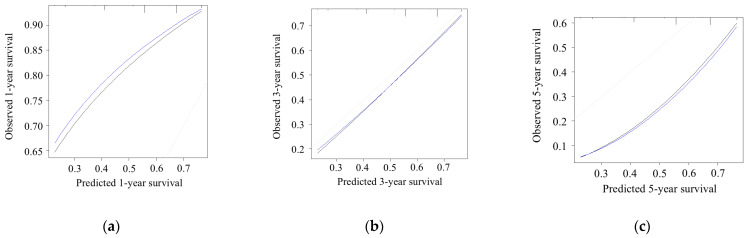
Calibration plots of the model for (**a**) 1-year, (**b**) 3-year, and (**c**) 5-year overall survival. The concordance index of the model was 0.63. Black line, observed; gray line, ideal; blue line, optimism corrected.

**Table 1 cancers-17-03694-t001:** Clinicopathological characteristics of the 603 patients.

Variables	*n* = 603
Preoperative factors	
Age, years	72 (65–77)
Sex, male/female	325 (53.9%)/278 (46.1%)
BMI, kg/m^2^	21.9 (19.7–24.0)
Biliary drainage	210 (34.8%)
CA19-9, U/mL	111 (30–392)
Alb, g/dL	4.0 (3.6–4.3)
Total bilirubin, mg/dL	0.8 (0.6–1.3)
CRP, mg/dL	0.1 (0.1–0.4)
mGPS, 0/1/2	418 (69.3%)/137 (22.7%)/48 (8.0%)
Radiological findings	
Tumor location, head/body-tail	393 (65.2%)/210 (34.8%)
Tumor size, cm	2.3 (1.8–3.0)
Tumor contact with the PV&SMV of ≤180°	174 (28.9%)
Regional lymph node metastasis	99 (16.4%)
Preoperative clinical findings (UICC/AJCC 7th edition)	
cT status, T1/T2/T3	29 (4.8%)/26 (4.3%)/548 (90.9%)
cStage, IA&B/IIA/IIB	53 (8.6%)/483 (80.1%)/68 (11.3%)
Operative factors	
Surgical procedure, PD/DP/TP	387 (64.2%)/198 (32.8%)/18 (3.0%)
PV&SMV resection	132 (21.9%)
Operative time, min	407 (284–521)
Estimated blood loss, mL	420 (220–730)
Pathological findings (UICC/AJCC 7th edition)	
pT status, T1/T2/T3	48 (8.0%)/31 (5.1%)/524 (86.9%)
Lymph node metastasis	331 (54.9%)
PV&SMV invasion	135 (22.4%)
R status, R0/R1 resection	532 (88.2%)/71 (11.8%)
pStage, IA&B/IIA/IIB	66 (10.9%)/204 (33.8%)/333 (55.2%)
Postoperative factors	
Major complications (≥grade 3)	135 (22.4%)
Pancreatic fistula (≥grade B)	114 (18.9%)
Adjuvant chemotherapy	452 (75.0%)
Recurrence	381 (63.2%)

Values are reported as *n* (%) or median (interquartile range). CA 19-9, carbohydrate antigen 19-9; ALB, albumin; CRP, C-reactive protein; mGPS, modified Glasgow Prognostic Score; PV&SMV, portal and superior mesenteric veins; PD, pancreatoduodenectomy; DP, distal pancreatectomy; TP, total pancreatectomy.

**Table 2 cancers-17-03694-t002:** Uni- and multivariate Cox regression analysis of preoperative prognostic factors for overall survival after pancreatectomy.

Preoperative factors	*n* (%)	Univariate Analysis		Multivariate Analysis	
		HR (95% CI)	*p* Value	HR (95% CI)	*p* Value
Age, year					
≥75	227 (37.6%)	1.20 (0.95, 1.52)	0.119		
<75	376 (62.4%)	Ref			
Sex					
Female	278 (46.1%)	0.88 (0.70, 1.10)	0.880		
male	325 (53.9%)	Ref			
BMI, kg/m^2^					
≥25	114 (18.9%)	1.12 (0.84, 1.48)	0.427		
<25	489 (81.1%)	Ref			
Biliary drainage					
Yes	210 (34.8%)	1.44 (1.14, 1.81)	0.002	1.17 (0.89, 1.55)	0.251
No	393 (65.2%)	Ref		Ref	
Tumor location					
Body-tail	210 (34.8%)	0.78 (0.61, 1.00)	0.046	1.00 (0.75, 1.35)	0.986
Head	393 (65.2%)	Ref		Ref	
Tumor size, cm					
>2.0	364 (60.4%)	1.74 (1.37, 2.22)	<0.001	1.50 (1.17, 1.93)	0.001
≤2.0	239 (39.6%)	Ref		Ref	
Tumor contact with the PV&SMV of ≤180°					
Presence	174 (28.9%)	1.54 (1.21, 1.95)	<0.001	1.47 (1.14, 1.88)	0.003
Absence	429 (71.1%)	Ref		Ref	
CA19-9, U/mL					
≥500	129 (21.4%)	2.58 (1.86, 3.58)	<0.001	2.16 (1.55, 3.04)	<0.001
40–500	299 (49.6%)	1.64 (1.23, 2.20)	<0.001	1.59 (1.19, 2.14)	0.002
<40	175 (29.0%)	Ref		Ref	
mGPS					
2	48 (8.0%)	1.73 (1.15, 2.50)	0.009	1.56 (1.03, 2.29)	0.038
0, 1	555 (92.0%)	Ref		Ref	

HR, hazard ratio; CI, confidence interval; BMI, body mass index; PV&SMV, portal and superior mesenteric veins; CA 19-9, carbohydrate antigen 19-9; mGPS, modified Glasgow Prognostic Score.

**Table 3 cancers-17-03694-t003:** Clinicopathological characteristics stratified by the risk groups.

Variables	Low-Risk *n* = 240	Moderate-Risk *n* = 313	High-Risk *n* = 50	*p* Value
Preoperative factors				
CA19-9, U/mL	29 (12–103)	198 (74–565)	1032 (699–3029)	<0.001
Alb, g/dL	4.1 (3.8–4.3)	4.0 (3.6–4.2)	3.7 (3.2–4.1)	<0.001
mGPS, 0/1/2	180 (75.0%)/57 (23.8%)/3 (1.2%)	216 (69.0%)/72 (23.0%)/25 (8.0%)	22 (44.0%)/8 (16.0%)/20 (40.0%)	<0.001
Radiological findings				
Tumor location, head/body-tail	136 (56.7%)/104 (43.3%)	212 (67.7%)/101 (32.3%)	45 (90.0%)/5 (10.0%)	<0.001
Tumor size, cm	1.8 (1.4–2.0)	2.5 (2.2–3.2)	3.0 (2.6–3.5)	<0.001
Tumor contact with the PV&SMV of ≤180°	24 (10.0%)	112 (35.8%)	38 (76.0%)	<0.001
Regional lymph node metastasis	22 (9.2%)	63 (20.1%)	14 (28.0%)	<0.001
Operative factors				
Surgical procedure, PD/DP/TP	133 (55.4%)/100 (41.7%)/7 (2.9%)	213 (68.1%)/94 (30.0%)/6 (1.9%)	41 (82.0%)/4 (8.0%)/5 (10.0%)	<0.001
PV&SMV resection	24 (10.0%)	83 (26.5%)	25 (50.0%)	<0.001
Operative time, min	349 (254–493)	422 (307–519)	495 (422–573)	<0.001
Estimated blood loss, mL	390 (200–676)	450 (235–797)	561 (393–826)	0.004
Pathological findings (UICC 7th edition)				
pT status, T1/T2/T3	39 (16.3%)/18 (7.5%)/183 (76.3%)	9 (2.9%)/12 (3.8%)/292 (93.3%)	0 (0%)/1 (2.0%)/49 (98.0%)	<0.001
Lymph node metastasis	94 (39.2%)	200 (63.9%)	37 (74.0%)	<0.001
PV&SMV invasion	34 (14.2%)	81 (26.0%)	20 (40.0%)	<0.001
R status, R0/R1 resection	216 (90.0%)/24 (10.0%)	272 (86.9%)/41 (13.1%)	44 (88.0%)/6 (12.0%)	0.53
pStage, IA&B/IIA/IIB	52 (21.7%)/92 (38.3%)/96 (40.0%)	13 (4.2%)/100 (32.0%)/200 (63.9%)	1 (2.0%)/12 (24.0%)/37 (74.0%)	<0.001
Postoperative factors				
Major complications (≥grade 3)	59 (24.6%)	64 (20.5%)	12 (24.0%)	0.49
Pancreatic fistula (≥grade B)	55 (22.9%)	54 (17.3%)	5 (10.0%)	0.05
Adjuvant chemotherapy	172 (71.7%)	244 (78.0%)	36 (72.0%)	0.21
Recurrence	115(47.9%)	224 (71.6%)	42 (84.0%)	<0.001

Values are reported as n (%) or median (interquartile range). CA 19-9, carbohydrate antigen 19-9; ALB, albumin; mGPS, modified Glasgow Prognostic Score; PV&SMV, portal and superior mesenteric veins; PD, pancreatoduodenectomy; DP, distal pancreatectomy; TP, total pancreatectomy.

**Table 4 cancers-17-03694-t004:** A summary of studies reporting nomograms for predicting survival after surgery.

Authors	Year	Study Design	Sample Size	Outcomes	Parameters	C-Index
Brennan et al. [25]	2004	Single-center, resected PDAC	555	CSS	Age Sex Weight loss Back pain pT stage pN stage Tumor size Differentiation Tumor location Margin status Splenectomy Portal vein resection	0.64
Attiyeh et al. [26]	2018	Single-center, resected PDAC	161: Training (*n* = 113) Testing (*n* = 48)	OS	Model A CA 19-9 Image features Model B CA 19-9 Image features Brennan score	Model A 0.68 Model B 0.73
He et al. [27]	2018	SEER database, resected pancreatic head cancer	2374: Training (*n* = 1780) Validation (*n* = 594)	OS CSS	Age Tumor size Differentiation Lymph node ratio M factor	0.64 (OS) 0.65 (CSS)
Li et al. [28]	2020	SEER database, resected PDAC	6323: Training (*n* = 3700) Validation (*n* = 1312) Prospective test (*n* = 1311)	OS CSS	Age Sex pT stage pN stage M stage Marital status Differentiation Tumor size Lymph node count Lymph node ratio	0.64 (OS) 0.64 (CSS)
Ren et al. [9]	2020	Single-center, resected PDAC	368: Training (*n* = 258) Validation (*n* = 110)	OS	CA19-9 pT stage pN stage Differentiation Capsule invasion Neutrophil percentage Transfusion Adjuvant therapy	0.77 (5-year OS)
Peng et al. [29]	2021	Single-center, resected pancreatic head cancer	307: Training (*n* = 307) External validation (*n* = 159)	OS	Age CA 19-9 Tongji classification pT stage pN stage Differentiation Adjuvant chemotherapy	0.79
Zou et al. [30]	2021	SEER database, resected pancreatic head cancer	6419: Training (*n* = 4495) Validation (*n* = 1924)	OS	pT stage pN stage Differentiation Adjuvant radiotherapy Adjuvant chemotherapy	0.68
Zhuang et al. [8]	2021	Single-center, resected pancreatic head cancer	112	OS	Age CA19-9 Total bilirubin pAJCC stage 8th Perineural invasion	0.73
Yoon et al. [7]	2022	Multicenter, resected PDAC	885	OS	Age Underlying liver disease Chronic kidney disease Preoperative portal vein invasion Portal vein resection Blood loss Tumor size Differentiation Lymph node metastasis Tumor necrosis	NA
Our study	2025	Multicenter, resectable PDAC	603	OS	Tumor size Tumor contact with the PV&SMV of ≤180° CA 19-9 mGPS	0.63

PDAC, pancreatic ductal adenocarcinoma; CSS, cancer-specific survival; OS, overall survival; CA 19-9, carbohydrate antigen 19-9; SEER, Surveillance, Epidemiology, and End Results; AJCC, American Joint Committee on Cancer; NA, not available; PV&SMV, portal and superior mesenteric veins; mGPS, modified Glasgow Prognostic Score.

## Data Availability

The original contributions presented in this study are included in the article. Further inquiries can be directed to the corresponding authors.

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
