# Peer review of "Prognosis Prediction Model After Upfront Surgery for Resectable Pancreatic Ductal Adenocarcinoma: A Multicenter Study (OS-HBP-2)"

_cancers, 2025, doi:10.3390/cancers17223694_

Round 1
Reviewer 1 Report
Comments and Suggestions for Authors
Comments
This multicenter retrospective study evaluates preoperative prognostic factors and proposes a prognostic prediction model for patients undergoing upfront surgery for resectable PDAC. The sample size is adequate, the methodology is generally appropriate, and the topic is clinically relevant. This work addresses an important clinical issue and provides useful stratification for patients with resectable PDAC. After the following suggested revisions and clarifications, the manuscript will be considerably strengthened.
1. Model performance and clinical applicability
The discriminative performance of the prognostic model (AUC 0.67–0.68; C-index 0.63) indicates only moderate predictive ability. To better contextualize the novelty of this work, a direct comparison with previously published nomograms or risk models (e.g., as a supplementary Table or Figure) is strongly recommended.
2. Clarification of CA19-9 cutoffs and confounding variables
Although the cutoffs (40 and 500 U/mL) are reasonable, CA19-9 levels can be significantly influenced by biliary obstruction. Please provide additional justification or consider adjustment/sensitivity analyses regarding biliary drainage status to reinforce the robustness of the results.
3. Implications for treatment optimization
As the moderate- and high-risk groups demonstrated substantially worse outcomes, the clinical relevance for decision-making regarding not only neoadjuvant therapy but also intensified adjuvant therapy should be further elaborated to enhance the practical utility of the risk stratification.
Author Response
Reviewer #1
This multicenter retrospective study evaluates preoperative prognostic factors and proposes a prognostic prediction model for patients undergoing upfront surgery for resectable PDAC. The sample size is adequate, the methodology is generally appropriate, and the topic is clinically relevant. This work addresses an important clinical issue and provides useful stratification for patients with resectable PDAC. After the following suggested revisions and clarifications, the manuscript will be considerably strengthened.
Comment 1:
Model performance and clinical applicability
The discriminative performance of the prognostic model (AUC 0.67–0.68; C-index 0.63) indicates only moderate predictive ability. To better contextualize the novelty of this work, a direct comparison with previously published nomograms or risk models (e.g., as a supplementary Table or Figure) is strongly recommended.
Response 1:
Thank you for your feedback. In the revised manuscript, we have performed a PubMed literature search to identify articles that developed nomograms to predict overall survival after surgery in patients with PDAC (pages 9-10, lines 232-242). Table 4 summarizes the ten included articles, including our study [7-9,25-30]. The study design included single-center (n = 5), multicenter (n = 2), and database (n = 3) studies. Previously published nomograms are targeted for resected PDAC. The C-index ranged from 0.63 to 0.79.
 In addition, we have discussed several novel findings of our model (pages 12-13, lines 302-309). As previous models targeted resected PDAC, our model focused only on patients with resectable PDAC. The prognostic factors for survival differ depending on the resectability status. Therefore, the resectability-specific prediction model was reasonable. Moreover, existing nomograms include postoperative parameters such as pathological findings that cannot be assessed preoperatively. In contrast, our model included only preoperative factors. Therefore, we could estimate patient survival preoperatively and construct treatment strategies depending on the risk categories.
Comment 2:
Clarification of CA19-9 cutoffs and confounding variables
Although the cutoffs (40 and 500 U/mL) are reasonable, CA19-9 levels can be significantly influenced by biliary obstruction. Please provide additional justification or consider adjustment/sensitivity analyses regarding biliary drainage status to reinforce the robustness of the results.
Response 2:
Thank you for your feedback. As CA19-9 levels can be significantly influenced by biliary obstruction, we selected preoperative CA19-9 levels after biliary drainage in this study. Of the 603 patients, 210 (34.8%) received biliary drainage. Median CA19-9 level was 111 U/mL (IQR 30–392 U/mL), and preoperative total bilirubin was within the normal range of 0.8 mg/dL (IQR 0.6–1.3 mg/dL). In the revised manuscript, we have provided this information (page 3, lines 117-119; page 4, lines 151-153).
Comment 3:
Implications for treatment optimization
As the moderate- and high-risk groups demonstrated substantially worse outcomes, the clinical relevance for decision-making regarding not only neoadjuvant therapy but also intensified adjuvant therapy should be further elaborated to enhance the practical utility of the risk stratification.
Response 3:
Thank you for your feedback. The potential clinical implications of the model would include survival risk stratification for patients with resectable PDAC. As shown in Table 3 and Figure 2, the OS and recurrence rates in the moderate- and high-risk groups were significantly lower than those in the low-risk group. Our results suggest that resectable PDAC should not be categorized solely by resectability status but rather subcategorized using various clinical factors, and appropriate multidisciplinary treatments, including neoadjuvant and adjuvant chemotherapy, should be provided. Although the role of neoadjuvant chemotherapy in resectable PDAC remains controversial, our prognostic prediction models may be useful for identifying high-risk groups that may benefit from intensified multidisciplinary care. Considering the growing evidence for neoadjuvant treatments for resectable PDAC, neoadjuvant chemotherapy may improve oncological outcomes and survival, especially in patients with high-risk features of resectable PDAC [38-40]. Moreover, evidence for adjuvant therapy for PDAC is well established [23,41-43]. The primary rationale behind the use of neoadjuvant and adjuvant therapies in pancreatic cancer is to target and destroy occult micrometastatic disease, which is believed to be the leading cause of early and frequent disease recurrence after curative surgery [44]. Accordingly, a perfect clinical course, including neoadjuvant chemotherapy, surgery, and adjuvant therapy, may be the ultimate protocol for improving outcomes, especially in moderate- and high-risk groups. In the revised manuscript, we have discussed this issue (page 12, lines 283-301).
Reviewer 2 Report
Comments and Suggestions for Authors
well written paper looking at predictive scoring for patients for upfront surgery in Pancreatic cancer
couple of small points
- the authors do not define R1 and R0 resections. It would be useful if they could give the definition used in this study as there is controversy as to whether simply the margin edge being free or whether the margin should be 0.5,1 , 1.5 or 2 mm. It would be useful to know what margin distance was used for this margin. It should also be made clear whether it was based on the closest margin for all resection margins or only for 1 particular margin
- The authors refer to the pstage of the patients. This is not fully defined but i presume they are using the 7th ed AJCC staging manual. This should be clarified
- In the discussion they refer to poor prognosis due to incomplete resection and early post operative recurrence. This should be expanded into discussion of early recurrence secondary to micrometastatic disease which has not been detected on staging which is the rationale behind adjuvent and neoadjuvent regimes to attempt to destroy these cell groups
Author Response
Reviewer #2
well written paper looking at predictive scoring for patients for upfront surgery in Pancreatic cancer
couple of small points
Comment 1:
The authors do not define R1 and R0 resections. It would be useful if they could give the definition used in this study as there is controversy as to whether simply the margin edge being free or whether the margin should be 0.5,1 , 1.5 or 2 mm. It would be useful to know what margin distance was used for this margin. It should also be made clear whether it was based on the closest margin for all resection margins or only for 1 particular margin.
Response 1:
Thank you for your feedback. The definition of R0 resection in this study included the margin edge being free based on the closest margin among all resection margins. In the revised manuscript, we have described the definition as suggested (page 3, lines 120-121).
Comment 2:
The authors refer to the pstage of the patients. This is not fully defined but i presume they are using the 7th ed AJCC staging manual. This should be clarified.
Response 2:
Thank you for your feedback. As the reviewer 2 pointed out, Pathological stages of the patients were defined using the criteria from AJCC-7th. In the revised manuscript, we have described the definition as suggested (page 3, lines 119-120).
Comment 3:
In the discussion they refer to poor prognosis due to incomplete resection and early post operative recurrence. This should be expanded into discussion of early recurrence secondary to micrometastatic disease which has not been detected on staging which is the rationale behind adjuvent and neoadjuvent regimes to attempt to destroy these cell groups.
Response 3:
Thank you for your feedback. As the reviewer 2 pointed out, the primary rationale behind the use of neoadjuvant and adjuvant therapies in pancreatic cancer is to target and destroy occult micrometastatic disease, which is believed to be the leading cause of early and frequent disease recurrence after curative surgery. In the revised manuscript, we have discussed this issue (page 12, lines 296-301).
Reviewer 3 Report
Comments and Suggestions for Authors
Dear Authors,
The study addresses an important clinical issue, given the poor survival rates of PDAC and the need for better prognostic tools. The multicenter design and large sample size enhance the generalizability of the findings. The methods are clearly described, and the statistical analyses are appropriate for the objectives. The development and internal validation of the prediction model are well-documented. The manuscript itself is well-written and provides valuable insights into the prognostic factors for survival after upfront surgery in patients with resectable PDAC. With some minor revisions, the manuscript would be suitable for publication.
Overall, the study is well-conducted and addresses an important issue. However, the lack of external validation and the moderate performance of the prediction model are limitations that need to be addressed. In addition, the retrospective nature of the study may introduce selection bias and limit the causality of the findings. The lack of external validation for the prediction model is a significant limitation. External validation is crucial to ensure the model's applicability in different populations. The discriminative performance of the model, while satisfactory, is not exceptional (AUC of 0.68 and C-index of 0.63). The authors should discuss how this model compares to existing predictive tools in terms of performance.
Comments:
- The authors should provide more context on how their prediction model compares to existing models. Are there any advantages or novel aspects of their model that make it more useful or accurate than previous ones?
- The discussion of the limitations should include a more detailed explanation of how the lack of external validation might affect the model's clinical utility.
- Please provide a more detailed comparison with existing predictive models to highlight the advantages or novel aspects of their model.
- Discuss the potential clinical implications of their findings in more detail.
Author Response
Reviewer #3
The study addresses an important clinical issue, given the poor survival rates of PDAC and the need for better prognostic tools. The multicenter design and large sample size enhance the generalizability of the findings. The methods are clearly described, and the statistical analyses are appropriate for the objectives. The development and internal validation of the prediction model are well-documented. The manuscript itself is well-written and provides valuable insights into the prognostic factors for survival after upfront surgery in patients with resectable PDAC. With some minor revisions, the manuscript would be suitable for publication.
Overall, the study is well-conducted and addresses an important issue. However, the lack of external validation and the moderate performance of the prediction model are limitations that need to be addressed. In addition, the retrospective nature of the study may introduce selection bias and limit the causality of the findings. The lack of external validation for the prediction model is a significant limitation. External validation is crucial to ensure the model's applicability in different populations. The discriminative performance of the model, while satisfactory, is not exceptional (AUC of 0.68 and C-index of 0.63). The authors should discuss how this model compares to existing predictive tools in terms of performance.
Comment 1:
The authors should provide more context on how their prediction model compares to existing models. Are there any advantages or novel aspects of their model that make it more useful or accurate than previous ones?
Response 1:
Thank you for your feedback. In the revised manuscript, we have performed a PubMed literature search to identify articles that developed nomograms to predict overall survival after surgery in patients with PDAC (pages 9-10, lines 232-243). Table 4 summarizes the ten included articles, including our study [7-9,25-30]. The study design included single-center (n = 5), multicenter (n = 2), and database (n = 3) studies. Previously published nomograms are targeted for resected PDAC. The C-index ranged from 0.63 to 0.79.
 In addition, we have discussed several novel findings of our model (pages 12-13, lines 302-309). As previous models targeted resected PDAC, our model focused only on patients with resectable PDAC. The prognostic factors for survival differ depending on the resectability status. Therefore, the resectability-specific prediction model was reasonable. Moreover, existing nomograms include postoperative parameters such as pathological findings that cannot be assessed preoperatively. In contrast, our model included only preoperative factors. Therefore, we could estimate patient survival preoperatively and construct treatment strategies depending on the risk categories.
Comment 2:
The discussion of the limitations should include a more detailed explanation of how the lack of external validation might affect the model's clinical utility.
Response 2:
Thank you for your feedback. As the reviewer 3 pointed out, its retrospective nature may have introduced a potential selection bias in surgical deci-sion-making across centers and limited the causality of the findings. Using multivariate analysis, we identified independent preoperative predictors associated with OS. However, other confounding or residual confounding factors may exist. The discrimi-native performance of the prognostic prediction model was investigated using internal validation. However, the moderate performance of the prediction model may be a lim-itation. Moreover, an external validation was not performed to verify the reliability and generalizability of the model. Because external validation is crucial to ensure the mod-el's applicability in different populations, the lack of external validation of the predic-tion model is a significant limitation. Future studies should conduct external validations to confirm the applicability of this model. In the revised manuscript, we have discussed our limitations in more detail (page 13, lines 310-321).
Comment 3:
Please provide a more detailed comparison with existing predictive models to highlight the advantages or novel aspects of their model.
Response 3:
Thank you for your feedback. In the revised manuscript, we have performed a PubMed literature search to identify articles that developed nomograms to predict overall survival after surgery in patients with PDAC (pages 9-10, lines 232-243). Table 4 summarizes the ten included articles, including our study [7-9,25-30]. The study design included single-center (n = 5), multicenter (n = 2), and database (n = 3) studies. Previously published nomograms are targeted for resected PDAC. The C-index ranged from 0.63 to 0.79.
 In addition, we have discussed several novel findings of our model (pages 12-13, lines 302-309). As previous models targeted resected PDAC, our model focused only on patients with resectable PDAC. The prognostic factors for survival differ depending on the resectability status. Therefore, the resectability-specific prediction model was reasonable. Moreover, existing nomograms include postoperative parameters such as pathological findings that cannot be assessed preoperatively. In contrast, our model included only preoperative factors. Therefore, we could estimate patient survival preoperatively and construct treatment strategies depending on the risk categories.
Comment 4:
Discuss the potential clinical implications of their findings in more detail.
Response 4:
Thank you for your feedback. The potential clinical implications of the model would include survival risk stratification for patients with resectable PDAC. As shown in Table 3 and Figure 2, the OS and recurrence rates in the moderate- and high-risk groups were significantly lower than those in the low-risk group. Our results suggest that resectable PDAC should not be categorized solely by resectability status but rather subcategorized using various clinical factors, and appropriate multidisciplinary treatments, including neoadjuvant and adjuvant chemotherapy, should be provided. Although the role of neoadjuvant chemotherapy in resectable PDAC remains controversial, our prognostic prediction models may be useful for identifying high-risk groups that may benefit from intensified multidisciplinary care. Considering the growing evidence for neoadjuvant treatments for resectable PDAC, neoadjuvant chemotherapy may improve oncological outcomes and survival, especially in patients with high-risk features of resectable PDAC [38-40]. Moreover, evidence for adjuvant therapy for PDAC is well established [23,41-43]. The primary rationale behind the use of neoadjuvant and adjuvant therapies in pancreatic cancer is to target and destroy occult micrometastatic disease, which is believed to be the leading cause of early and frequent disease recurrence after curative surgery [44]. Accordingly, a perfect clinical course, including neoadjuvant chemotherapy, surgery, and adjuvant therapy, may be the ultimate protocol for improving outcomes, especially in moderate- and high-risk groups. In the revised manuscript, we have discussed this issue (page 12, lines 285-301).
Round 2
Reviewer 1 Report
Comments and Suggestions for Authors
The authors have responded accurately and completely to all of my requests.
As a result, the manuscript is now well-structured and has become even more solid and improved. Thank you very much.
Reviewer 2 Report
Comments and Suggestions for Authors
The authors have answered and addressed the points i raised to my satisfaction and i would recommend publication